# *Trypanosoma cruzi* Antigenic Proteins Shared with Acute Lymphoblastic Leukemia and Neuroblastoma

**DOI:** 10.3390/ph15111421

**Published:** 2022-11-17

**Authors:** Leticia Eligio García, María del Pilar Crisóstomo Vázquez, Víctor Alberto Maravelez Acosta, Mariana Soria Guerrero, Adrián Cortés Campos, Enedina Jiménez Cardoso

**Affiliations:** Laboratorio de Investigación en Parasitología, Hospital Infantil de México Federico Gómez (HIMFG), Dr. Márquez 162. Col Doctores, Cuauhtémoc, México City 06720, Mexico

**Keywords:** acute lymphoblastic leukemia, *Trypanosoma cruzi*, neuroblastoma, antigens, immunotherapy

## Abstract

**Background.** Research studies indicate that immunization with protein extracts of *Trypanosoma cruzi*, the protozoan parasite that causes Chagas disease, prevents the appearance of tumors in 60% of mice injected with the murine lung carcinoma tumor line. The molecular basis of this process is unknown, although the presence of specific antigens in tumor cells and on the surface of *T. cruzi* suggests an antiparasitic immune response, with an effective cross-reaction against cancer cells, hence the importance to identify the antigens involved and determine their potential as target cells in anticancer therapy. **Aim.** This study aimed to determine the presence of antigenic proteins of *T. cruzi* shared with acute lymphoblastic leukemia and neuroblastoma cells. **Material and methods.** To achieve this, polyclonal antibodies against *T. cruzi* were developed in rabbits, and reactivity was determined with protein extracts of acute lymphoblastic leukemia cells and neuroblastoma. The immunodetection of five different strains of *T. cruzi* against anti-*T. cruzi* polyclonal antibodies was also performed. **Conclusion.** The study allows the knowledge of the immunological interactions between cancer and parasites to be expanded and, therefore, contributes to the design of more and better projects that improve the therapeutic strategies applied in oncology.

## 1. Introduction

Cancer is a generic term that designates a wide group of diseases that can affect any part of the body. A defining characteristic of cancer is the rapid multiplication of abnormal cells that spread beyond their normal limits and can invade adjacent parts of the body or spread to other organs. This process, called metastasis, is the leading cause of cancer death [1]. Although incidence rates for all cancers combined are twice in more developed countries with respect to less developed ones, death rates are only 8% to 15% higher in developed countries due to risk factors, screening practices, and availability of treatment [2]. In Mexico, cancer is the third leading cause of death, 14/100 Mexicans die from this disease and 45.3% of deaths occur in the economically active population [3].

Acute lymphoblastic leukemia (ALL) is a malignant transformation and proliferation of lymphoid progenitor cells in the bone marrow, blood, and extramedullary sites. Overall, 80% occurs in children and it represents a devastating disease when it occurs in adults. The incidence of ALL follows a bimodal distribution, with the first peak in childhood and the second peak around 50 years old [4]. Symptoms of ALL include enlarged lymph nodes, bruising, fever, bone pain, frequent infections, and bleeding gums. Treatment may include chemotherapy or local-release drugs that kill cancer cells. Although dose escalation strategies have led to a significant improvement in outcomes for pediatric patients, the prognosis for the elderly remains extremely poor [5].

Neuroblastoma (NB) is an embryonic autonomic nervous system tumor usually found in the adrenal glands [6]. It can develop in the stomach, chest, neck, pelvis, and bones. It affects children under five years of age, and it is the most common cancer diagnosed during the first year of life. The most frequent symptoms are fatigue, loss of appetite, and fever. The clinical presentation is highly variable, from a mass that does not cause symptoms to a primary tumor that causes critical illness because of local invasion, widely disseminated disease, or both [7].

In this work, it was decided to work with ALL and NB because they are two of the most frequently diagnosed in children and adults, respectively, and because of the lack of studies to recognize antigens shared between these oncological cells and *Trypanosoma. cruzi*.

Parasitic infections caused by protozoa and helminths induce chronic inflammation that, added to cell divisions of the healing process, favors the appearance of cancer [8,9,10]. However, there is scientific evidence that some species could help in the fight against tumors. The design of effective vaccines for the treatment of cancer is one of the main challenges in cancer research. Several nonpathogenic parasites, such as *Leishmania tarentolae*, *Toxoplasma gondii*, and *T. cruzi*, have been used as candidates for designing cancer vaccines [11]. There is growing experimental and clinical evidence that the immune system is actively involved in the pathogenesis and control of tumor progression. An effective antitumor response depends on the correct interaction of various components of the immune system, such as antigen-presenting cells and different subpopulations of T cells. However, malignant tumors develop efficient mechanisms to evade their recognition and elimination. The beneficial effects reported for parasitic diseases in tumorigenesis range from the induction of apoptosis, the activation of the immune response, the avoidance of metastasis and angiogenesis, and the inhibition of proliferative signals to the regulation of inflammatory responses that promote cancer [12]. It has been reported for the parasitic malaria infection that inhibits cell proliferation in lung cancer [13] or by intratumoral injection of attenuated *T. gondii* for cases of melanoma [14]. It was also shown that, when administered to mice, three peptides from *E. granulosus* confer splenocytes with the ability to kill tumor cells. Furthermore, this cytotoxic activity was correlated with a higher number of activated NKs, which suggests that the immune response may be efficient to eliminate tumor cells in the presence of this infection [15]. *E. granulosus* is a cestode parasite that causes cystic echinococcosis disease. In a retrospective study, a significantly lower prevalence of cancer was seen in patients with hydatid disease [16]. Antigenic similarities were found between *E. granulosus* and some types of tumors [17]. As well as this, immunization with human hydatid cystic fluid (HCF) induces antibodies against CT26 colon carcinoma cells and protects against tumor growth in mice [18].

It has also been proven that *T. cruzi*, the protozoan parasite that causes Chagas disease, has mediated anticancer effects, with by-products derived from the parasite that inhibit the growth of tumor cells, involving both the cellular and humoral components of the immune response. Therefore, it is proposed to develop polyclonal antibodies against *T. cruzi* in rabbits and to determine the reactivity with NB tumor protein extracts and with protein extract of ALL cells in culture to determine the presence of the molecules involved in this process. The objective of this study was to determine the presence of antigenic proteins of *T. cruzi* shared with protein extract of cells in culture of ALL and extract protein of NB.

## 2. Results and Discussion

*T. cruzi*, the protozoan parasite that causes Chagas disease, shows anticancer effects through the presence of proteins that inhibit tumor cell growth [19]. There are scientific reports that show various molecular targets have the antitumor ability. However, it has not been determined which is the molecul with anticancer activity, nor the process that is carried out. The studies have been conducted in cell cultures and experimental animals and suggest the following facts: *T. cruzi* has antitumor effects by inducing host immunity against tumors. *T. cruzi* expresses calreticulin that can directly interact with endothelial cells and inhibit their proliferation, migration, and capillary morphogenesis, as well as inhibit tumor cells [20,21].

### 2.1. Determination of Antibodies by ELISA

The results obtained from ELISA immunodetection are shown in Figure 1. The protein extract of ALL culture was used as a target antigen. Subsequently, purified sera from hyperimmunized rabbits with the five different isolates of *T. cruzi* were used as the first antibody, and anti-rabbit IgG horseradish peroxidase (HRP) enzyme conjugate as a detection antibody. The substrate for the enzyme was added to quantify the primary antibody through a color change. The OD values were read at 450 nm with a reference filter of 620 nm in a spectrophotometer. The ALL control had an Ab titer of 2.674 and, for *T. cruzi*, it was from 1.228 (Ninoa) to 2.664 (Cuernavaca).

The immunization of laboratory animals with total extracts of *T. cruzi* strains allowed sera with a high antibody titer to be obtained, determined by ELISA (greater than 1.6); however, it is observed that the immunogenicity is different between them since the Silvio and Ninoa isolates present a lower test titer (Figure 1). A dot blot was performed to ensure that there was an immunogenic reaction between the immunized serum and the protein extract of the ALL cells in culture and, although the detection signal was weak, it ensured the presence of polyclonal antibodies, which could be used in the Western blot method.

### 2.2. SDS Polyacrylamide Gel Electrophoresis

The recognition by Western blotting of the ALL and NB antigens (total extract) shows different specific bands for each isolate. Figure 2 shows the electrophoretic patterns of cell extract ALL and NB, respectively; both were run by 12% SDS PAGE under the same running conditions.

### 2.3. Immunodetection 

The anti-*T. cruzi* antibodies were used to detect the presence of proteins shared with the electrophoresed cells of ALL and NB. In the case of ALL, with the strains CLB, Cuernavaca, and Querétaro, while, with NB, antigenic proteins were immunodetected in all strains (Figure 3). The homologous immunodetection of Ab was made by Western blot using the protein extract of *T. cruzi* strain as antigen and immunodetected with Ab anti- *T. cruzi* of each of five strains of *T. cruzi* (Figure 4).

Figure 2 shows the different electrophoretic patterns of a cell extract of ALL and NB, writing down the presence of proteins that vary in size and pH and are the basis for the later performance of Western blot. The results of the Western blot (Figure 3) show that *T. cruzi* do share antigenic molecules with the ALL cells in culture and the NB extract developed in a mouse, which was recognized by specific antibodies present in serum. In the case of ALL, the recognition was with antibodies from CLB, Queretaro, and Cuernavaca strains, where the recognition of fragments of 95, 75, 70, 55, 50, and 43 KDa is seen. With the antibodies against Silvio and Ninoa strains, they did not show recognition, so they are not included in the figure. Regarding NB, recognition is observed with the immunized sera of the five strains, showing bands of 140, 90, 85, 80, 70, and 55 KDa. The bands recognized by the antibodies were observed with different intensities according to the heterogeneity of concentration in the cell extract. In this case, there was a reaction with the five strains, unlike ALL, which suggests that the genetic plasticity of *T. cruzi* also influences the recognition of shared antigenic proteins.

This plasticity is evidenced in Figure 4, where immunized serum against *T. cruzi* isolates recognized antigens in homologous protein extracts and, among them, the presence of different antigens is observed, which may be the origin of the differences between strains when recognizing antigens of ALL and NB. Reported research says that the antitumor capacity of *T. cruzi* is influenced by the discrete typing unit (DTU) to which the strain corresponds [22].

The heterologous immunodetection shows that there are shared antigens between the extracts of ALL and NB cells and *T. cruzi* cells and, on the other hand, the homologous immunodetection shows that the protein plasticity between strains of *T. cruzi* is remarkably high, since the antibodies obtained from each one by immunization in rabbits shows significant differences. Although the presence of *T. cruzi* and other infectious organisms have been reported to be involved in carcinogenesis, some of them with antitumor capacity have also been reported [23]. This study is a contribution to the knowledge of proteins involved in the mechanism that causes *T. cruzi* proteins to induce a decrease in oncological processes since the specific mechanisms of action are still not clear.

### 2.4. Confocal Microscopy

Figure 5 shows the result of the confocal microscopy assay with ALL and NB cells against anti-*T. cruzi* antibodies, using DAPI and Alexa 488 as fluorogenic markers. Rabbit pre-immune serum was used as a negative control and *T. cruzi* cells were used to show the specificity of the antibodies.

Confocal microscopy assay showed the presence of antigenic proteins by immunodetection in ALL and NB cells with antibodies against *T. cruzi*. Recognition is more intense in NB cells than in ALL cells. These results show the shared proteins of *T. cruzi* with ALL and NB cells. Therefore, it sets up the need for further research to find which are the most important and how they participate in the anticancer capacity of Trypanosoma. This study suggests that infection by *T. cruzi* generates a powerful immune response capable of enhancing and directing an efficient immune response against oncological cells. *T. cruzi* antigens have common epitopes with mammalian mucins [24] and with glycoproteins that share sialyl-Tn-like structures (specific human structures associated with cancer) [25]. We think all researchers are necessary to carry out studies of the antitumor action of *T. cruzi* and their products for the discovery and use of new molecules that may be proposed as therapeutic alternatives in the treatment of cancer.

## 3. Materials and Methods

### 3.1. Production and Determination of Antibodies against T. cruzi in New Zealand Rabbits

A hyperimmune anti-*T. cruzi* serum was attempted by immunizing a male New Zealand rabbit (about 6 weeks old). Blood was drawn before the first immunization, and this was the pre-immune serum used as a control. It was administered subcutaneously with 300 μg of total extract of *T. cruzi* epimastigotes (axenic culture) strain CL Brener in 500 μL of complete Freund’s adjuvant; 15 days later 300 μg of total extract was administered with 500 μL of incomplete Freund’s adjuvant. Later, on days 29, 30, and 31, 100 μg of total extract dissolved in a saline solution were administered intravenously; finally, on day 39, venous blood was taken from the rabbit, which was centrifuged to separate the serum, which was collected, aliquoted, and stored at −20 °C. The total IgG antibodies present in the serum were purified from the polyclonal serum generated in a rabbit. For this, a protein A bound to agarose (22811, Thermo Fischer Scientific, Waltham, MA, USA) was used. The anti-*T. cruzi* antibody titer was obtained in the serum of the immunized rabbits by ELISA. The OD values were read at 450 nm with a reference filter of 620 nm in an ELISA reader (ELISA LKB Wallac Olivetti AU). Each assay plate included negative and positive samples as controls [26].

#### 3.1.1. Parasite Culture

Four different isolates of *T. cruzi* from infected humans in Mexico: Ninoa, Silvio, Queretaro, Cuernavaca, and strain CLB, were used to obtain specific antibodies of different isolates. Epimastigotes of four isolates and CLB control strain of *T. cruzi* were cultured and kept at 28 °C in 50 mL Nunc boxes of liver infusion tryptose (LIT) medium pH 7.2. After a period of 8 days, approximately 1 × 108 parasites/mL were obtained [27].

#### 3.1.2. Cellular Line 

ALL line SUP-B15 (RRID: CVCL_0103) was purchased and maintained in RPMI-1640 supplemented with 10% fetal bovine serum (FBS) and 1× streptomycin/penicillin antibiotics [28] and the NB line SH-SY5Y (RRID: CVCL_0019) was kept in Dulbecco’s modified Eagle’s medium (DMEM medium, Gibco, Grand Island, NY, USA), supplemented with 10% of FBS [29]. All experiments were performed with mycoplasma-free cells.

#### 3.1.3. Antigen Preparation

The parasites of isolates of *T. cruzi* were harvested in 50 mL Falcon tubes, the medium was removed by centrifugation at 2000 rpm for 20 min, the supernatant was removed, and two washes were carried out with saline solution at 1500 rpm for 5 min. Parasites were counted in a Neubauer chamber. The button formed by parasites was resuspended in 1 mL of lysis buffer with protease inhibitors and cell disruption was performed by vortexing for 5 min, followed by three freeze/thaw cycles [30].

#### 3.1.4. Cell Extract of ALL 

The extract was prepared from cultures of ALL Line Sup B15 in the exponential growth phase, washed 2 times with PBS, and subsequently incubated in a hypotonic solution (10 mM Tris, 5 mM KCl, 2 mM CaCl_2_, 1 mM MgCl_2_) supplemented with protease inhibitors at a final concentration of 500 mM EDTA, 20 mM PMSF, 10 mM leupeptin, and 1 mM pepstatin for 30 min on ice. The samples were sonicated with 6 pulses of 10 s, the material obtained was centrifuged at 4600 rpm for 10 min, and the supernatant was stored at −20 °C until use. The protein concentration of the parasitic extracts and the cell lines used was measured by spectrophotometry [31].

### 3.2. SDS Polyacrylamide Gel Electrophoresis

A dot blot method was conducted with 1 mg/mL of Ag from *T. cruzi* isolates and immunodetected with the first antibody (inoculated rabbit serum) diluted 1:50. Protein extract was treated with bromophenol blue and then heated to boiling for 15 min, thus achieving a more stable union between the SDS and the protein. Treated antigens (from *T. cruzi* or ALL proteins) were run in a polyacrylamide gel electrophoresis 12% for 40 min at 200 V.

### 3.3. Immunodetection

Electrophoresed proteins were transferred to a 0.2 µM nitrocellulose membrane (Trans-Blot-Transfer Medium Bio-Rad) for 3 h at 350 mA, 120 V in transfer buffer. The membrane was blocked overnight at 4 °C with 5% milk PBS, the strips of the membrane of approximately 1 cm were cut, the serum of the anti-*T. cruzi* rabbits diluted 1:50 in milk PBS was added to the 1% and incubated for 2 h at 37 °C, and positive controls were used to validate this test. After the incubation time, three washes with 0.1% PBS Tween were carried out, and the second anti-rabbit IgG antibody coupled to horseradish peroxidase was added in a 1:1000 dilution with 1% milk PBS, after which seven washes with PBS 0.1% Tween were performed. The strips were developed by adding the substrate 4 Chloro Naphthol (SIGMA); the reaction was carried out at room temperature for approximately 30 min and stopped with distilled water [32,33].

### 3.4. Confocal Microscopy

Sub15 ALL cells were cultured in RPMI 1640 medium (supplemented with 10% FBS and 1× antibiotic/antimycotic) and NB cells in DMEM medium. The axenic culture of *T. cruzi* was used as a positive control. Cells were harvested and washed with PBS. A total of 50,000 viable cells were fixed with 2% paraformaldehyde for 15 min. After 40 min of washing with PBS, it was blocked with 2% pig serum in PBS for 1 h and the permeabilized slides were incubated with 2% pig serum and 0.5% Triton in PBS for 20 min. They were incubated for 1 h with *T. cruzi* antibodies 1:300 with 2% pig serum in PBS. IgG purified from unimmunized New Zealand rabbit serum was used as a negative control. Reactivity was carried out with specific anti-rabbit IgG antibodies conjugated to Alexa-Fluor 488 (Invitrogen cat no. A-11034) diluted 1:300 with 2% pig serum in PBS for 1 h. Cell nuclei were stained with 20 µg/mL of 4,’6-diamidino-2-phenylindole dilactate (DAPI) (Invitrogen cat no. D1306). The images were taken with a Leica model TCS-5P8X confocal microscope, with a 63× immersion lens, 5× digital zoom, and analyzed with the LasX software.

## Figures and Tables

**Figure 1 pharmaceuticals-15-01421-f001:**
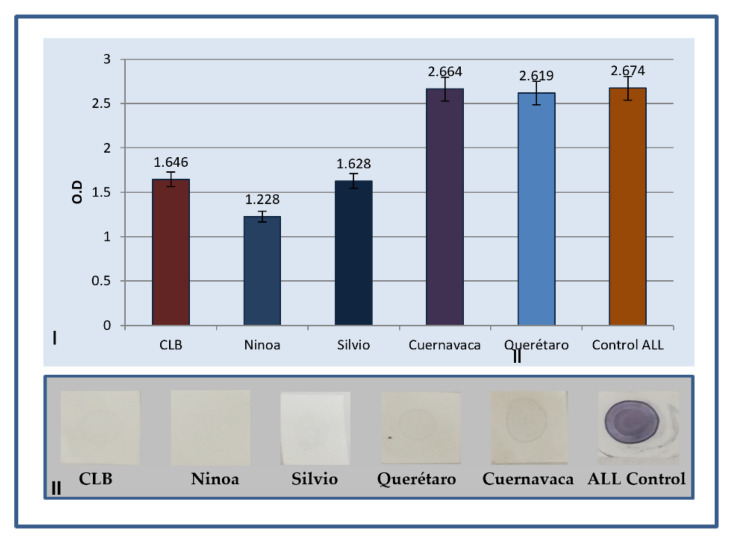
(**I**) Representation of antibody levels in the serum of rabbits immunized with the protein extract of *T. cruzi* isolates. Results are expressed as mean ± IC of the optical density (OD) at 520 nm measured in ELISA: (**II**) evaluation of ALL antigen vs. anti-*T. cruzi* polyclonal serum of five strains by dot blot.

**Figure 2 pharmaceuticals-15-01421-f002:**
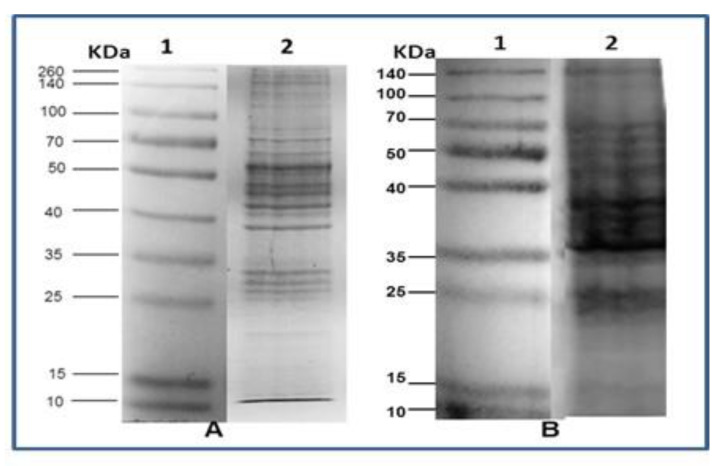
The result of 12% SDS-PAGE gel show lysate cell of (**A**) ALL and (**B**) NB. 1. Protein molecular weight marker and 2. electrophoretic pattern of 300 µg of protein extract.

**Figure 3 pharmaceuticals-15-01421-f003:**
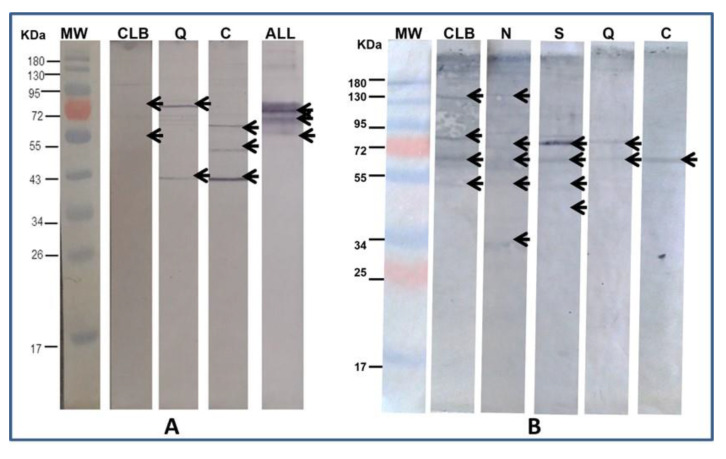
Western blot analysis using as antigen: (**A**) ALL cell lysate and (**B**) NB cell lysate vs. immunized serum against different isolates of *T. cruzi* (heterologous). MW.—protein molecular weight marker. CLB.—anti-CLB serum. N.—anti-Ninoa serum. S.—anti-Silvio serum. Q.—anti-Querétaro serum. C.—anti-Cuernavaca serum. ALL.—control anti-ALL serum. Arrows indicate the most significant bands that are shared between the different strains of *T. cruzi* with ALL and NB.

**Figure 4 pharmaceuticals-15-01421-f004:**
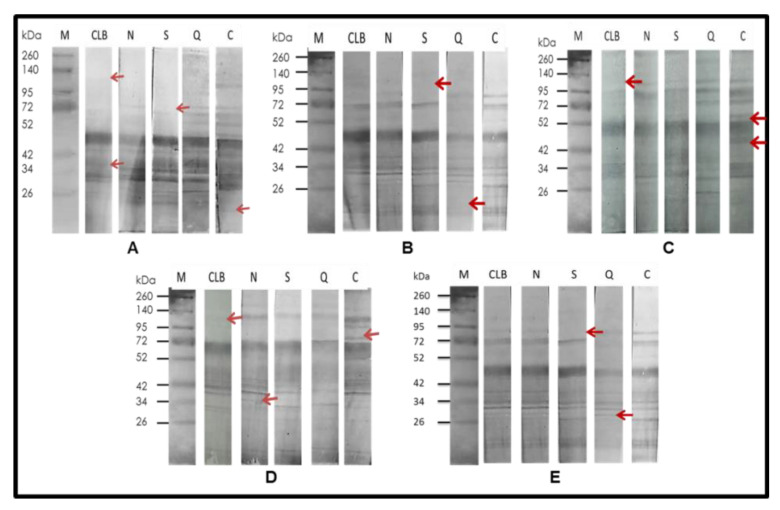
Western blot analysis: cell lysates of each strain of *T. cruzi* were electrophoresed by 12% SDS-PAGE and transferred onto nitrocellulose membrane, and then treated with antibodies against *T. cruzi* strains (Homologous). (**A**) CLB vs. CLBrenner (CLB), Ninoa (N), Silvio (S), Querétaro (Q), and Cuernavaca (C); (**B**) Ninoa vs. CLB, N, S, Q, and C; (**C**) Silvio vs. CLB, N, S, Q, and C; (**D**) Queretaro vs. CLB, N, S, Q, and C; (**E**) Cuernavaca vs. CLB, N, S, Q, and C. Arrows indicate differences observed between strains.

**Figure 5 pharmaceuticals-15-01421-f005:**
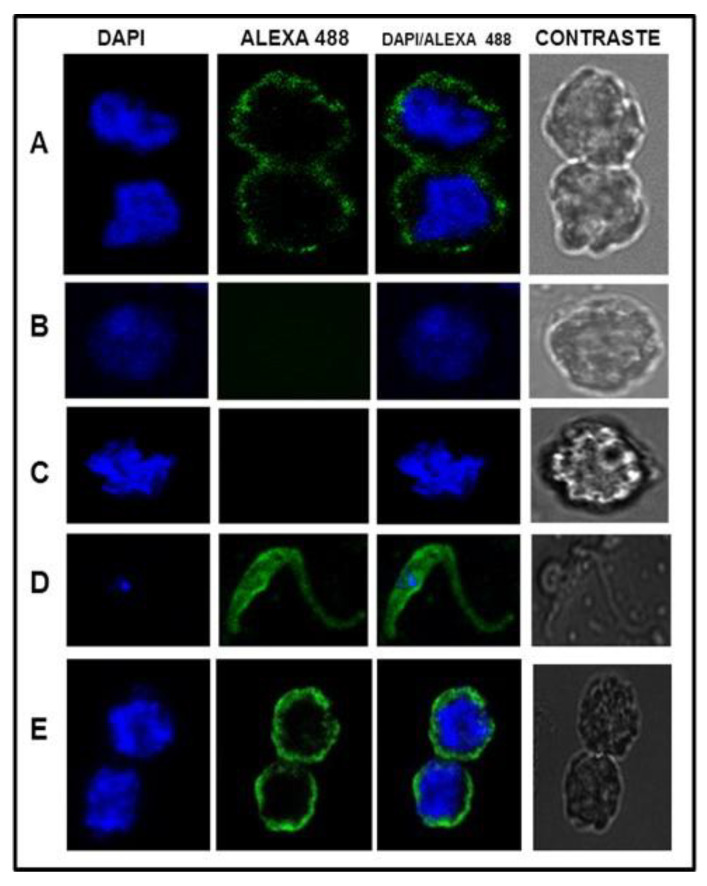
Confocal microscope analysis results from ALL and NB cells processed with immunofluorescence using polyclonal anti-*T. cruzi* sera. Assays with different fluorescent markers are shown. (**A**) ALL cells vs. anti-*T. cruzi* antibodies; (**B**) healthy leukocytes vs. anti-*T. cruzi* antibodies; (**C**) ALL cells vs. preimmunized serum; (**D**) cultured epimastigotes of *T. cruzi* vs. anti-*T. cruzi* antibodies; (**E**) NB cell vs. anti-*T. cruzi* antibodies. Scale bar = 10 µm.

## Data Availability

Data is contained within the article.

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
