# Peer review of "Trypanosoma cruzi Antigenic Proteins Shared with Acute Lymphoblastic Leukemia and Neuroblastoma"

_pharmaceuticals, 2022, doi:10.3390/ph15111421_

Round 1

Reviewer 1 Report

In this work, the presence of antigenic proteins of T.cruzi, shared with cells in culture of ALL and NB were determined. Polyclonal antibodies against T.cruzi were developed in rabbits, and the reactivity with protein extracts of cultured ALL and NB cells was investigated. In general, the paper is difficult to read, and the results hard to understand. The results and discussion must be joined, and more information should be added to better understand the different steps of the work.  English language and style must be improved. In this sense, I recommend major corrections before the manuscript is accepted for publication. Moreover, I suggest other modifications to improve the paper quality.

Abstract:

1.      Line 15: “Research studies indicate that immunization with protein extracts of T.cruzi, the…”. T.cruzi should be changed to Trypanosoma cruzi.

2.      Line 21. ALL e NB must be substituted to Acute lymphoblastic leukemia (ALL) and neuroblastoma (NB).

3.      Line 21. “The objective of this work is to determine the presence of antigenic proteins of T.cruzi, shared with cells in culture of ALL and NB, for this, polyclonal antibodies against T.cruzi were developed in rabbits, and the reactivity with protein extracts of cultured ALL and NB cells was determined, Likewise, the protein immunodetection of the different strains of T.cruzi was carried out with the anti-T.cruzi antibodies of the five strains.” This sentence must be divided in three distinct sentences.

Keywords: “Acute Lymphoblastic Leukemia, Trypanosoma, Antigens, immunotherapy, Neuroblastoma.” I Think Neuroblastoma should appear near Acute Lymphoblastic Leukemia.

Introduction: The introduction should be rewritten and the English improved, and specific issues changed.

1.      Line 31: “Cancer is a generic term that designates a wide group of diseases that can affect any part of the body; A defining characteristic of cancer is the rapid multiplication of abnormal cells that spread beyond their normal limits and can invade adjacent parts of the body or 33 spread to other organs, a process called metastasis, which is the leading cause of cancer death [1].” The “;” must be substituted by “.” and two sentences will be generated.

2.      Line 50: “Neuroblastoma is an embryonic tumor of the autonomic nervous system; it is a type of cancer that is usually found in the adrenal glands”. Change the “;” to “and”.

3.      Line 57: “It is historically known that parasites, both protozoa and helminths, can form longterm infections in humans whose chronic inflammation causes cancer. Some examples are Schistosoma mansoni, which causes malignant tumors [8], Plasmodium falciparum, which induces Burkitt's lymphoma [9] and Trichomonas vaginalis, which has been associated with 60 an increase in prostate cancer [10].” I think this sentence should be deleted.

4.      Line 93: “The aim of this study was determining the presence of antigenic proteins of T.cruzi, shared with protein extract of cells in culture of ALL and extract protein of NB.” This sentence must be integrated into the previously paragraph (line 86-92) since describes the work proposed.

Results: The results and discussion must be joined to easier understand what was done and the obtained results. In general, the results have to be better described and some issues must be clear, as:

1.      The results section must be reorganized, and subsections must be created. For example, 2.1 Determination of Antibodies by ELISA; 2.2 Immunoblot, etc…

2.      Line 96: “The results obtained when using the protein extract of ALL to measure the antibody titer by the ELISA method using the sera of rabbits immunized with the 5 different isolates of T.cruzi are shown in table 1.” This sentence is not clear. The protein extract of ALL is from where? The goal is to measure which antibodies in rabbit sera? More information has to be added to better understand this assay.

3.      Figure 1. Table II is not needed since the graph I has the same values. The graph must be edited to be self-explained. The image III has to have the strain instead of “A”, “B”….

Materials and Methods

1.      Line 230: “5mM, CaCl2 2mM, MgCl2 1mM supplemented with” Correct the subscripts.

Author Response

Consulte el archivo adjunto

Reviewer 2 Report

The article by Leticia Eligio García et al. entitled “Trypanosoma cruzi antigenic proteins, shared with acute lymphoblastic leukemia and neuroblastoma” was submitted to Pharmaceutics. The reviewer's enthusiasm remains limited due to the following concerns.

 1. Significant grammar and typographical errors were found throughout the manuscript. Authors from non-English speaking countries should ensure to have their articles corrected by a native English speaker for grammatical, stylistic, and typographical errors.

2. The author must rewrite the abstract with the background, the aim of the study, materials and methods, results, and conclusion clearly in the abstract without any extensive sentences

3. Abstract: What is the abbreviation of ALL, NB in the abstract, if using it the first time

4. Introduction: What is the role of the review related to Echinococcus granulosus (Line 80-85), and is it linked to Trypanosoma cruzi? Instead, authors should write an in-depth review of T. cruzi and its antigenic similarities.

5. Among the cancers, why did the authors focus on ALL and NB culture cells?

6. In order to validate the consistency of antigenic properties of T. cruzi, researchers can perform at least two cell lines for each cancer type.

7. Authors did not perform a murine model to support their hypothesis. Still, this work is primitive and needs to explore the underlying mechanism

Round 2

Reviewer 2 Report

Accept in present form